# Lumos: Optimizing Live 360-degree Video Upstreaming via Spatial-Temporal Integrated Neural Enhancement

## ABSTRACT

As VR devices become increasingly prevalent, live 360-degree video has surged in popularity. However, current live 360-degree video systems heavily rely on uplink bandwidth to deliver high-quality live videos. Recent advancements in neural-enhanced streaming offer a promising solution to this limitation by leveraging server-side computation to conserve bandwidth. Nevertheless, these methods have primarily concentrated on neural enhancement within a single domain (either spatial or temporal), which may not adeptly adapt to diverse video scenarios and fluctuating bandwidth conditions. In this paper, we propose Lumos, a novel spatial-temporal integrated neural-enhanced live 360-degree video streaming system. To accommodate varied video scenarios, we devise a real-time Neural-enhanced Quality Prediction (NQP) model to predict the neural-enhanced quality for different video contents. To cope with varying bandwidth conditions, we design a Content-aware Bitrate Allocator, which dynamically allocates bitrates and selects an appropriate neural enhancement configuration based on the current bandwidth. Moreover, Lumos employs online learning to improve prediction performance and adjust resource utilization to optimize user quality of experience (QoE). Experimental results demonstrate that Lumos surpasses state-of-the-art neural-enhanced systems with an improvement of up to 0.022 in terms of SSIM, translating to an 8.2%-8.5% enhancement in QoE for live stream viewers.

## CCS CONCEPTS

• **Information systems** → **Multimedia streaming**; • **Networks** → **Network resources allocation**.

## KEYWORDS

live streaming, 360-degree video, neural enhancement, quality of experience

**ACM Reference Format:**
Anonymous Authors. 2024. Lumos: Optimizing Live 360-degree Video Upstreaming via Spatial-Temporal Integrated Neural Enhancement. In *Proceedings of the 32nd ACM International Conference on Multimedia (MM'24), October 28-November 1, 2024, Melbourne, Australia.* ACM, New York, NY, USA, 10 pages. https://doi.org/10.1145/nnnnnnn.nnnnnnn

## 1 INTRODUCTION

Live 360-degree video streaming has emerged as a significant portion of Internet traffic [41], offering users an exceptionally immersive experience [9]. Viewers can adjust their viewing angles freely, enabling exploration of omni-directional video content during live events. Recent market reports [29] indicate a consistent growth in demand for live 360-degree video streaming. On the upstream side, upload clients typically employ a 360-degree camera equipped with multiple lenses to capture live video streams. These streams are subsequently sliced into tiles, encoded using various codecs [3, 5, 23, 31, 36], and transmitted through protocols like HTTP, RTMP, as highlighted in [1, 25, 26], to the media server.

However, achieving high-quality live 360-degree video upstreaming encounters challenges due to constraints in uplink bandwidth [46, 47] and the computational capacity of the upload client. Limited uplink bandwidth necessitates the compression and transmission of video at lower quality, leading to poor video quality. Moreover, attaining high-quality live 360-degree video upstreaming mandates real-time 360-degree video encoding, posing challenges for mobile devices with limited computational capacity.

Recent advancements in neural-enhanced video streaming have introduced a novel approach to enhance video quality by applying neural computation to video frames [15, 16, 21]. The fundamental concept is to leverage the computational power of the media server to offset the limitations of uplink bandwidth. Broadly, these approaches can be categorized into two primary categories: *one* employs super-resolution (SR) techniques for live video enhancement in the spatial domain [14, 35, 44], while *the other* utilizes frame-interpolation (FI) techniques in the temporal domain [13, 24, 33]. However, downsampling in the spatial domain may compromise image details [19, 42], especially in frames with intricate textures, whereas downsampling in the temporal domain may adversely affect motion information in high-action videos [6, 11]. Consequently, relying solely on neural enhancement in either the spatial or temporal domain has inherent drawbacks that may result in a subpar user experience.

Nonetheless, the integration of spatial-temporal neural enhancement in live 360-degree video upstreaming introduces several new and non-trivial challenges. *Firstly*, the spatial-temporal combination necessitates the upload client to explore optimal neural enhancement configurations within an expanded solution space. However, due to the limited computational power of the upload client, rapidly searching for the optimal solution and performing neural enhancement is challenging. *Secondly*, video tiles in a live 360-degree stream exhibit different rate-distortions after encoding. Equally allocating bitrate among tiles would lead to inefficient bandwidth usage and degrade the quality of transmitted videos. Estimating the required bandwidth for each tile to optimize the rate-distortion poses a challenge. *Thirdly*, minimizing the degradation of user Quality of Experience (QoE) over time is challenging, as dynamic video

scenes limit the effectiveness of historical data in improving system performance. Moreover, constrained bandwidth makes it difficult to refresh the dataset, intensifying the complexity of the problem.

In this paper, we propose Lumos, a novel live 360-degree video streaming framework that integrates spatial and temporal domain neural enhancement techniques. By considering different ratios of downsampling in the spatial and temporal domains, Lumos generates a wide array of diverse neural enhancement configurations, each corresponding to varying levels of neural-enhanced quality. This enables Lumos to overcome the aforementioned limitations by exploring the optimal neural enhancement configuration in a two-dimensional solution space, a capability not feasible with existing methods limited to a single dimension.

To address the aforementioned challenges, Lumos incorporates several important innovative designs. *Firstly*, Lumos introduces a lightweight Neural-enhanced Quality Prediction (NQP) model trained using historical live 360-degree video data. This model takes the low-level spatial-temporal features of tile videos as input and generates upper and lower bounds of neural-enhanced quality predictions for various spatial-temporal neural enhancement configurations. This guidance aids in selecting optimal spatial-temporal configurations, reducing the computational burden on the upload client while enhancing the robustness of the configuration selection procedure. *Secondly*, based on our observation of a significant correlation between the spatial-temporal complexity of different tiles and their rate-distortions, Lumos adopts a dynamic bandwidth allocation strategy. This strategy assigns available bandwidth to each tile according to its specific spatial-temporal complexity. By doing so, Lumos ensures efficient bandwidth utilization, thereby facilitating the transmission of higher-quality live 360-degree videos. *Finally*, recognizing that even low-quality tiles can provide valuable labels for updating the dataset, Lumos utilizes the remaining bandwidth after transmitting 360-degree videos to send low-quality tiles for updating the NQP model. This proactive approach mitigates performance degradation and maintains user QoE.

In summary, our paper makes the following contributions:

- We conduct measurement experiments to explore the relationship between spatial-temporal complexity and tile rate-distortion, and find a strong correlation between them. Additionally, we observe that low-quality tiles can also effectively serve as training samples for updating neural models used in system components to maintain system performance.
- We propose Lumos, the first spatial-temporal integrated live 360 video streaming framework to our knowledge. In Lumos, we introduce a real-time NQP model to predict the neural-enhanced quality for various video contents. Additionally, we present a tile-level content-aware bitrate allocator that dynamically allocates bandwidth and selects appropriate neural enhancement configurations for tiles.
- We conduct extensive evaluations using real-world traces. The experimental results demonstrate that Lumos outperforms state-of-the-art methods with up to 0.022 SSIM gain and enhances user Quality of Experience (QoE) by 8.2%-8.5%.

The remainder of this paper is organized as follows. In Section 2, we survey previous works related to our research. In Section 3, we conduct the measurement studies to understand the drawbacks of one-dimension neural enhancement and discover the key observation inspiring our design. In Section 4, we introduce the detailed design of Lumos. In Section 5, we demonstrate experiments results of Lumos compared with other baselines. Finally, we conclude our work in Section 6.

## 2 RELATED WORK

### 2.1 360-Degree Live Video Streaming

Previous works on the optimisation of 360-degree video streaming systems have been approached from various perspectives. Yi *et al.* [45] conducted a detailed measurement study on the time consumption of a live 360-degree video streaming system and discovered the relationship between task time consumption and system latency. Feng *et al.* [10] developed a viewport prediction scheme for live 360-degree video streaming systems using motion tracking and user interest modelling. Ban *et al.* [2] proposed a multi-agent deep reinforcement learning based 360-degree video streaming system (MA360) to solve the multi-user streaming problem in edge caching networks. Sun *et al.* [32] proposed the use of "flocking" to improve viewport prediction and edge server caching for live 360-degree video streaming. Xie *et al.* [37] proposed a dynamic co-rendering solution for low-latency and high-quality mobile virtual reality. Eltobgy *et al.* [8] proposed a solution for live 360-degree video multicast to mobile users.

### 2.2 Neural-enhanced Video Streaming

**Super-resolution Based Neural Enhanced Video Streaming System.** Recent advances in deep learning, especially in super-resolution, offer opportunities to reduce bandwidth consumption in video streaming applications. Dasari *et al.* [7] proposed a super-resolution approach that significantly compresses the video on the server side and enhances it to higher quality on the client side using a neural network model. Kim *et al.* [14] introduced LiveNAS, a real-time video ingestion framework that enhances the quality of the original video stream by leveraging server-side computation. Live-NAS applies neural enhancement to the original video stream and uses online learning to maximize quality. Luo *et al.* [20] developed CrowdSR, a real-time video ingestion method that converts low-resolution video streams from weak devices into high-resolution streams through super-resolution. Wang *et al.* [35] proposed Vaser, a novel neural-enhanced 360-degree live video ingestion framework considering viewport information. While these works focus on the spatial domain, recovering original details and ensuring video quality for frames with complex textures can be challenging. **Frame-interpolation Based Neural Enhanced Video Streaming System.** Some existing works optimize video transmission by discarding packets or video frames. Yahia *et al.* [39] discard fixed bitrate video frames in live video streams using HTTP/2. Stewart *et al.* [30] modify the SCTP protocol to be partially reliable, allowing for discarding some packets without considering video frame characteristics. BETA [13] considers all B-frames as discardable, while VOXEL [24] modifies P-frames and B-frames for unreliable transmissions. Yang *et al.* [40] design an efficient end-to-end collaborative VR streaming system to increase the frame rate of VR video, but it is not easily applicable to real-time video streaming systems. Reparo [33] propose a new approach for real-time video

streaming to improve QoE in low-rate networks, using a lightweight prediction model to determine whether to drop an even frame and recovering it through VFI at the media server. These works provide inspiration for compressing video in the time domain, but none of them handle video content with drastically changing scenes effectively.

# 3 MEASUREMENT STUDY

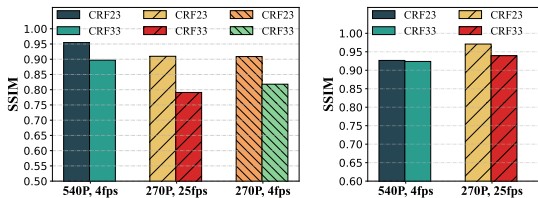

(a) Neural-enhanced Quality on Complex Textured Tile.

(b) Neural-enhanced Quality on High Dynamic Tile.

Figure 1: The enhanced quality of different neural enhancement configurations on different types of tiles.

**Inherent drawbacks of one-dimension neural enhancement.** In a typical neural-enhanced live streaming video system, the video is usually downsampled either spatially or temporally to further compress its size, followed by neural enhancement using SR-DNN and FI-DNN.

Figure 1a and Figure 1b illustrate the performance of different neural enhancement configurations on different types of tiles. We showcase the effects of neural enhancement using three different neural enhancement configurations: (a) *temporal downsampling only* (540P, 4fps), (b) *spatial downsampling only* (270P, 25fps), and (c) *spatial-temporal downsampling* (270P, 4fps). *CRF* is an FFmpeg encoding option used for bitrate control, where the larger the CRF, the lower the bitrate of the encoded video. As shown in Figure 1a, the DNN-based SR model falls short of effectively enhancing video tile with complex textures. Furthermore, simultaneous spatial-temporal downsampling yields comparable neural enhancement results with a smaller tile size. However, as shown in Figure 1b, temporal downsampling does harm to video quality in scenarios with strong dynamics, suggesting that neural enhancement in the temporal dimension is not applicable to these tiles.

These results suggest that neural enhancement in either the spatial or temporal domain has its own drawbacks, and an integration of the two should be considered in practice to improve video quality. **Correlation between rate-distortion and spatial-temporal complexity.** In 360-degree video transmission systems, adaptive bandwidth allocation for each tile is essential. As illustrated in Figure 2a and 2b, the Tile 0 achieves remarkably high quality with less bandwidth requirement, while the Tile 12 requires a higher bandwidth allocation to achieve comparable quality, indicating the rate-distortion varies across different tiles. What's more, we observed a strong correlation between the tile's rate-distortion and its spatial-temporal complexity, as depicted in Figure 2a. To achieve satisfactory video quality, a tile with high spatial-temporal complexity requires a much higher bitrate than a tile with low

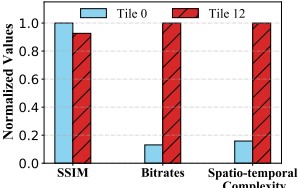
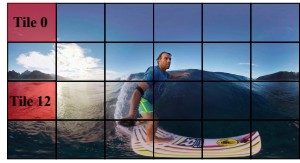

(a) The normalized values of SSIM, bitrates and spatial-temporal complexity.

(b) The corresponding tiles in the 360-degree video corresponds to the left outcome.

Figure 2: The video quality, bitrates and spatial-temporal complexity for different tiles in the same chunk.

spatial-temporal complexity. This insight inspires us to estimate the rate-distortion based on the spatial-temporal complexity of tiles, enabling adaptive bandwidth allocation among tiles to achieve rational utilization of bandwidth.

**Effectiveness of low-quality training samples.** Due to the limited remaining bandwidth after video transmission, it is not feasible to transmit the original and complete video tiles for online updates of the NQP model. In our observations, we find out that tiles that are complete in resolution and frame rate but encoded at a very low bitrate have the potential to be used as retraining samples for the NQP model. As shown in Figure 3, except for the lower bound (*CRF33*) of the configuration *540p, 12fps*, the relative order of the upper and lower bounds of other configurations is the same, indicating their potentials as training labels. Moreover, the size of the low-quality samples is small enough to be transmitted using remaining bandwidth, showing their usefulness.

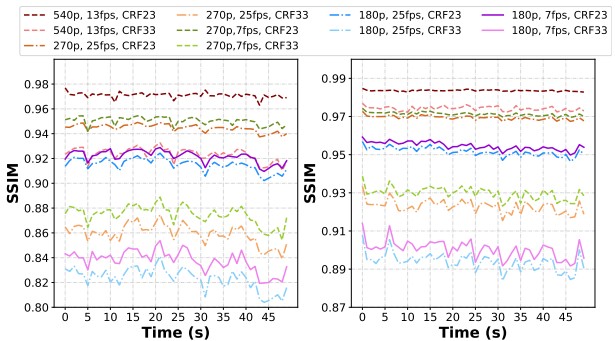

Figure 3: The enhanced quality bounds of 5 neural enhancement configurations on the origin tile frames (left) and low-quality tile frames (right).

# 4 FRAMEWORK DESIGN OF LUMOS

## 4.1 System Overview

The core of Lumos is to utilize spatial-temporal integrated neural enhancement techniques to make full use of constrained upload bandwidth, to transmit higher-quality 360-degree videos. Figure 4 elucidates the architectural framework of Lumos, which consists of an upload client and a media server.

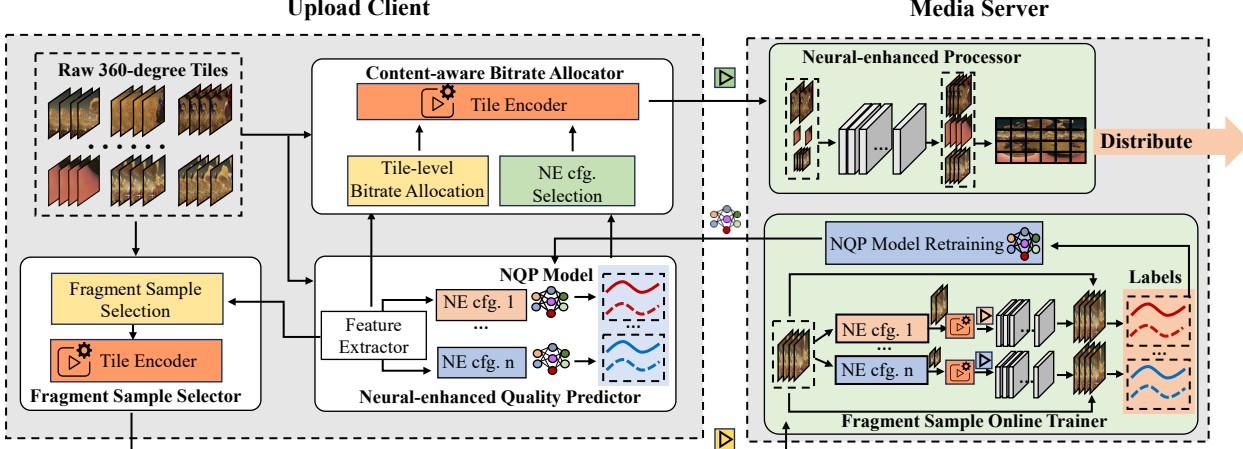

**Figure 4: Lumos system overview, where *NE cfg.* denotes neural enhancement configuration.**

**Upload Client.** At the upload client, Firstly, for each tile, the NQP model takes the low-level spatial-temporal feature as input and predicts the neural-enhanced quality of its corresponding neural enhancement configuration. Afterward, the available bitrates are dynamically allocated among tiles based on their spatial-temporal complexity, followed by selecting the appropriate neural enhancement configuration that satisfies the allocated bitrates. Then, the tile is encoded with the selected configuration and transmitted. Finally, leveraging the rest bandwidth after transmitting 360-degree videos, low-quality tiles, which are denoted as fragment sample in our design, are selected and sent to the media server for the NQP model updating.

**Media Server.** At the media server, firstly, the received 360-degree tile videos are decoded, and applied the corresponding spatial-temporal neural enhancement techniques for recovery. The enhanced tiles are then stitched and transcoded into multiple versions waiting for viewer requests. Simultaneously, fragment training samples are decoded into tile frames. For each neural enhancement configuration, the tile frames are downsampled, encoded, and decoded using the highest and lowest bitrate settings. After that, the corresponding neural enhancement configuration is applied to enhance the tile frames to obtain the upper and lower bounds of enhanced quality, serving as labels for re-training the NQP model. Then, the NQP model is updated and sent back to the upload client periodically.

## 4.2 Neural-enhanced Quality Prediction

The role of the Neural-enhanced Quality Predictor is to forecast the highest and lowest restore quality for each neural enhancement configuration.

**Extracting spatial-temporal features of tiles.** As depicted in Figure 4, firstly, the Neural-enhanced Quality Predictor extracts spatial-temporal features of each tile. Inspired by Reducto[17], the Feature Extractor is designed to capture a range of low-level features to construct the spatial-temporal feature for each tile, such as pixel difference and area difference for the temporal domain and edge difference and variance of pixel values after applying the Laplacian

operator [4] for the spatial domain. The spatial-temporal features of tiles are then utilized for predicting neural-enhanced quality, as well as for computing spatial-temporal complexity.

**Predicting neural-enhanced quality.** Based on the aforementioned steps, the NQP model is trained to predict the neural-enhanced quality for its corresponding neural enhancement configuration. This prediction serves as a guidance for the selection of the neural enhancement configuration for each tile.

At time chunk $i$, assume the spatial-temporal features of tile $j$ be denoted as $f_{ij}$ and $\text{NQP}_k$ denotes the NQP model of neural enhancement configuration $k$. For each configuration $k$, we predict the neural-enhanced quality for each tile, denoted as $q_{ij}^k$. This procedure could be established as follows:

$$q_{ij}^k = \text{NQP}_k(f_{ij}). \tag{1}$$

To ensure the effectiveness of our NQP model predictions, we propose two design strategies. First, for each neural enhancement configuration, rather than predicting the neural-enhanced quality under all CRF settings, we only forecast the upper and lower bounds of the enhanced quality. That is the neural-enhanced quality of tile videos which is encoded with corresponding neural enhancement configuration and the highest and lowest CRF. This approach not only reduces the number of prediction outputs to mitigate the risk of wrong selections but also affords greater flexibility in selecting spatial-temporal neural enhancement configurations, thereby enhancing the robustness of the proposed system. Furthermore, we employ lightweight Multilayer Perceptron (MLP) as the NQP model, which enables real-time inference under the constraint of limited computational resources.

## 4.3 Content-aware Bitrate Allocation

After predicting the neural-enhanced quality for each neural enhancement configuration, we proceed to allocate bitrates to each tile and select the appropriate configurations to downsample the tiles.

**Measuring spatial-temporal complexity.** Firstly, it is crucial to measure the spatial-temporal complexity of each tile. We employ the spatial-temporal feature to calculate the spatial-temporal complexity of each tile, as we aggregate all the low-level features within the spatial-temporal feature of each tile to serve as its spatial-temporal complexity.

---

**Algorithm 1:** ALLOCATE_BITRATES($B_i, z_{ij}, N$)

**input** : Spatial-temporal complexity for tile $j$ in chunk $i$: $z_{ij}$, avalible bitrates for chunk $i$: $B_i$, the numbel of tiles: $N$.

**output** : Allocated bitrate for tile $j$ in chunk $i$: $B_{ij}$.

1 *Initialise $w_{ij} \leftarrow \alpha$ for tile $j$ in chunk $i$* ;

2 $w_{ij} \leftarrow w_{ij} + z_{ij}, \forall j$;

3 $w_{ij} \leftarrow \frac{w_{ij}}{\sum_{j=1}^{N} w_{ij}}, \forall j$ ;

4 $w'_{ij} \leftarrow \text{TRUNCATE}(w_{ij}), \forall j$;

5 $w'_{ij} \leftarrow \frac{w_{ij}}{\sum_{j=1}^{N} w'_{ij}}, \forall j$;

6 $B_{ij} \leftarrow B_i \times w'_{ij}, \forall j$;

7 **return** $B_{ij}$;

---

**Allocating bitrate among tiles.** Then, bitrate allocation among tiles is conducted based on two considerations: first, tiles with lower spatial-temporal complexity should get lower bitrates as they can achieve satisfactory viewing quality with fewer bandwidth. Moreover, it is crucial not to allocate too much bandwidth to tiles with higher spatial-temporal complexity, as this may excessively degrade the quality of other tiles. Thus, a trade-off must be made between these two types of tiles to ensure full utilization of bandwidth and high-quality encoding of tiles.

We design a weight-based heuristic bitrate allocation algorithm reflecting the proportion of total bitrates allocated to a specific tile $j$ in chunk $i$. Initially, each tile is assigned an initial weight $\alpha$. Then, the weight of each tile is incremented by its corresponding spatial-temporal complexity $z_{ij}$, i.e., $w_{ij} = \alpha + z_{ij}$, where $w_{ij}$ is the weight of tile $j$ in chunk $i$. After that, the weight function is normalized to ensure that the total weight across all tiles sums up to one. To prevent the bitrate allocation from excessively favoring certain tiles and hurting others' quality, the normalized weights of each tile are then truncated to predefined maximum and minimum thresholds. Finally, the truncated weights are renormalized to obtain the bitrate weight for each tile, which is then used to proportionally allocate bitrates to each tile according to the formula in Algorithm 1.

**Selecting spatial-temporal downsampling configurations.** Given the predicted quality of each neural enhancement configuration and the allocated bitrates, we select the most suitable neural enhancement configuration for each tile. For each neural enhancement configuration, its upper and lower bounds of neural enhancement performance correspond to an encoding bitrate, respectively. For example, as shown in figure 5a, the upper and lower bounds of enhancement qualities for configuration **y** correspond to bitrates 60kbps and 40bps, respectively. Therefore, we follow these steps to select the neural enhancement configuration:

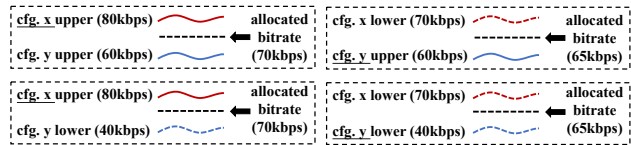

(a) Configurations selection according to the upper bound.
(b) Configurations selection according to the lower bound.

**Figure 5: Selecting neural enhancement configurations based on predicted quality and allocated bitrate, where the underlined neural enhancement configurations are the selected ones in that case.**

(1) First, we sort the predicted enhancement qualities (including upper and lower bounds) of each configuration in descending order.

(2) Second, we traverse through each configuration sequentially until we find one with a bitrate lower than the allocated bitrate for the tile (e.g. the configuration **y** in our examples).

(3) If the allocated bitrate lies between the upper bound of configuration **x** and the upper (or lower) bound of configuration **y** (where **y** can be any configuration, including **x**), we select configuration **x** (e.g., the example in Figure 5a); otherwise, if the allocated bitrate lies between the lower bound of configuration **x** and the upper (or lower) bound of configuration **y** (where **y** can be any configuration except **x** since the upper bound corresponds to a higher bitrate than the lower bound), we choose configuration **y** (e.g., the example in Figure 5b).

The above strategy is based on two considerations: (1) If the allocated bitrate falls between **the upper bound** of configuration **x** and the upper (or lower) bound of configuration **y**, it indicates that there exists a suitable CDF setting of configuration **x** that meets the requirements of the allocated bitrate, and selecting configuration **x** provides the opportunity for better neural-enhanced quality. (2) If the allocated bitrate falls between **the lower bound** of configuration **x** and the upper (or lower) bound of configuration **y**, it suggests that the allocated bitrate is not within the bitrate range of configuration **x**. In this case, we choose configuration **y** and select an appropriate CRF for tile encoding.

**Encoding tiles with selected neural enhancement configuration.** For each tile, we downsampled it to the resolution and frame rate corresponding to its selected neural enhancement configuration. The remaining tile frames are then encoded using a CDF that meets the allocated bitrate and transmitted to the media server side.

## 4.4 Fragment Sample Selection

The purpose of the Fragment Sample Selector is to effectively utilize the remaining bandwidth for transmitting as many fragment samples as possible for the online training of the NQP model. Therefore, we base the selection procedure on the following principles: (1) Regionalization: Adjacent tiles in a 360-degree video exhibit similar characteristics. (2) Importance: Tiles should be sorted by spatial-temporal complexity. (3) Balance: Uniform sampling is necessary while considering Importance. Therefore, samples with lower spatial-temporal complexity should also be collected.

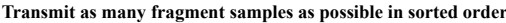

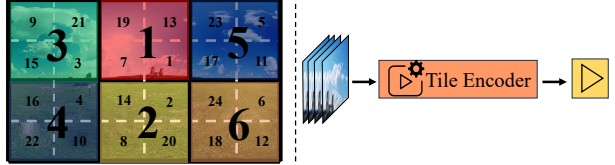

**Figure 6: Fragment sample transmission process, where large numbers represent regions and small numbers indicate the sorted order.**

We present the designed fragment sample selection procedure. Firstly, as illustrated in Figure 6, the 360-degree video is divided into six regions, labeled 1 to 6. Subsequently, tiles within regions 1 to 4 are sorted in descending order of spatial-temporal complexity, while tiles in regions 5 to 6 are sorted in ascending order. It is worth noting that due to the wrapping-around property of 360-degree video ERP projection, there is no left or right distinction in the video. Finally, following the sequence of regions 1 to 6, tiles are selected from each region according to the sorted order and encoded in low quality to become a fragment sample. These samples are then transmitted to the server side utilizing the remaining bandwidth until it is depleted.

### 4.5 Neural-enhanced Processing

The Neural-enhanced processor's role is to enhance the received downsampled 360-degree video tiles, restoring their original resolution and frame rate, and then stitching them together to form complete 360-degree video chunks.

**Enhancing received tiles with DNNs.** For tiles with reduced spatial resolution, we use SR-DNN to recover the resolution, while for tiles with reduced frame rate only, we use FI-DNN to recover the frame rate. When both the resolution and frame rate of the tile are degraded, we first use SR-DNN to recover the resolution and then apply FI-DNN to recover the frame rate. With the above process, we achieve spatial-temporal integrated neural enhancement for all 360-degree video tiles.

Subsequently, the neural-enhanced tiles are stitched together to form complete 360-degree video chunks and transcoded into versions of varying bitrates, awaiting download requests from viewers.

### 4.6 NQP Model Retraining

As the similarity between historical and current live 360-degree videos diminishes over time, it becomes necessary to conduct online training for the NQP model, enabling it to adapt to upcoming video content.

**Extracting training dataset.** Figure 4 illustrates the process of acquiring a dataset. For a fragment sample, we apply each neural enhancement configuration to it for spatial-temporal downsampling, and the downsampled tile frames are encoded using the maximum and minimum CRF settings. Then, the encoded tile is decoded and neural enhancement is applied accordingly to restore it to its original resolution and frame rate. Afterward, the restored tile is compared with the unprocessed tile to calculate the upper and lower bounds of the quality of the neural enhancement, which

are used as training labels for this neural enhancement configuration. It is worth noting that the SSIM values computed on the fragment sample may be different from those computed on the original tile frame, but we are only concerned with the relative order of the upper and lower enhanced quality bounds of the different neural enhancement configurations, which the fragment sample can provide.

The inputs corresponding to the extracted labels are the spatial-temporal features of the fragment sample tiles, which are merely floating-point numbers and directly transmitted from the upload client, occupying negligible bandwidth.

**Updating the NQP model.** Employing the described approach, the media server continuously updates the training dataset and periodically retrains the NQP model using the extracted data. Subsequently, it forwards the updated model to the upload client, replacing the obsolete NQP model.

## 5 EVALUATION

In this section, we evaluate Lumos with a series of experiments and in-depth analysis to answer the following questions:

(1) How much is the video quality enhancement of the proposed Lumos compared with other state-of-the-art systems that only consider a single dimension?

(2) Does Lumos offer improved Quality of Experience (QoE) for end users?

(3) Does the implementation of the Content-aware Bitrate Allocator and Fragment Sample Selector genuinely lead to performance improvement?

### 5.1 Experiment Settings

**Evaluation Setup.** The Lumos framework is implemented using Python, and a testbed is established for the data generation and evaluation process. Our pipeline operates on Intel(R) Xeon(R) CPU E5-2640 v4 @ 2.40GHz processors, complemented by NVIDIA GeForce RTX 3090 Super GPUs. For DNN training, both SR-DNN and FI-DNN are pre-trained on 360-degree videos from Gaze360 [38] that are not used for testing. We utilize the EDSR [18] model for spatial-domain neural enhancement and the RIFE [12] model for temporal-domain neural enhancement. These models are not updated online due to their acceptable generalization capabilities. In terms of low-level features, we choose four low-level features: pixel difference, area difference, edge difference, and the variance of pixel values after applying the Laplacian operator. Considering the NQP model, for each neural enhancement configuration, we train a simple MLP with three hidden layers of sizes 40, 20, and 2. The total size of the NQP model corresponding to all neural enhancement configurations is 31.79 KB, and it takes only 3 ms for inference on PICO4 [27], a VR headset with mobile-grade chips. Regarding the neural enhancement configurations, for each tile, spatially, we applied downsampling factors of [540P, 270P, 180P], and temporally, we applied downsampling factors of [25fps, 13fps, 7fps, 4fps]. It is important to note that for a fair comparison, we excluded the configuration [540P, 25fps] which does not undergo any neural enhancement operations, resulting in a total of 11 neural enhancement configurations.

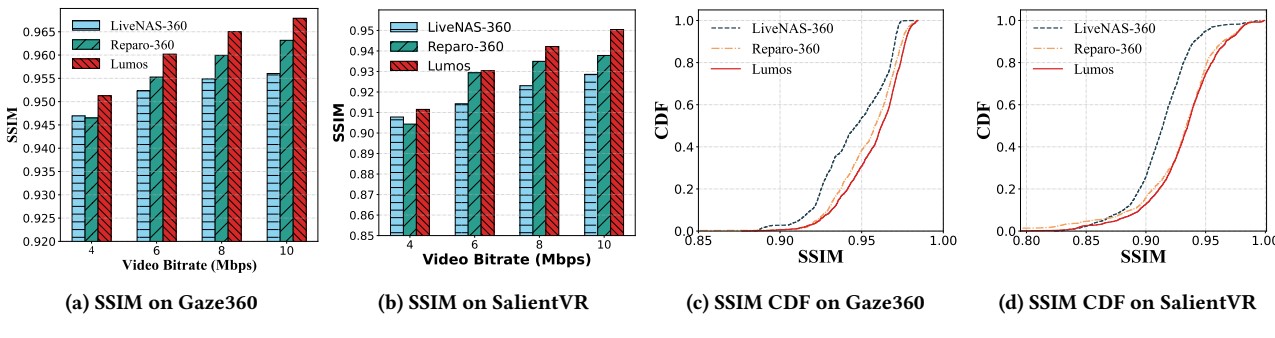

(a) SSIM on Gaze360     (b) SSIM on SalientVR     (c) SSIM CDF on Gaze360     (d) SSIM CDF on SalientVR

Figure 7: Video quality improvement of Lumos in two datasets.

**Evaluation Videos.** We conducted experiments using nine 360-degree videos from Gaze360 [38] and four 360-degree videos from SalientVR [34]. The original resolution of the 360-degree videos is 4K and the frame rate is 25fps. The videos are divided into 24 (4 rows x 6 columns) separate tiles, with each tile having an original resolution of 540P. The first 30 seconds of the videos are allocated for training the NQP model, while the remaining duration is used for video streaming simulation. For each tile, the videos are encoded using 11 CRF (Constant Rate Factor) values ranging from 23 to 33. Finally, the buffer length is set to 10s for the experiments.

**Network Traces.** To better evaluate the performance, we use real 4G network traces [28], containing 135 traces. In our experiments, we filter out the traces whose average bandwidth is higher than 10Mbps to model a bandwidth-constrained environment just like previous works [14, 33]. As a result, the network bandwidth ranged from 4 to 10 Mbps.

**QoE Calculation.** To assess the Quality of Experience (QoE), we utilize the linear QoE model introduced by Pensieve [22]:

$$QoE = \sum_{n=1}^{N} q(R_n) - \mu \sum_{n=1}^{N} T_n - \sum_{n=1}^{N-1} |q(R_{n+1} - q(R_n))|, \quad (2)$$

where $\mu$ is set to 4.3 like Pensieve, and the increase of SSIM is computed using its effective bitrate function, generated through linear interpolation similar to NAS [43].

**Baselines.** To demonstrate that Lumos can improve video quality and enhance users' QoE, we conduct comparisons between Lumos and two state-of-the-art neural-enhanced baseline methods:

- *LiveNAS-360* [14], which primarily emphasizes spatial neural enhancement. In this approach, the upload client adjusts the resolution of tiles based on the available uplink bandwidth, and the media server restores the resolution of these tiles using SR-DNN, which is the same as the SR-DNN used by Lumos. It is worth noting that LiveNAS performs online training on SR-DNNs while we do not, which is orthogonal to our work.
- *Reparo-360* [33], which emphasizes temporal neural enhancement. In this approach, the upload client utilizes the VFD model's prediction to discard even frames of tiles, and the media server restores the frame rate of these tiles using FI-DNN, which is the same as the FI-DNN used by Lumos. The VFD model is periodically updated on the server side and sent back to the client.

## 5.2 Experiment Results

**Video Quality Improvement.** Figures 7a and 7b present the average SSIM values of all video chunks from Gaze360 and SalientVR generated by Lumos and other baseline methods at four different bitrate levels. We have two main observations: Firstly, compared to LiveNAS-360 and Reparo-360, Lumos surpasses them in both datasets across all bitrate levels, with SSIM gains up to 0.012 and 0.005 on Gaze360 and up to 0.022 and 0.013 on SalientVR. Secondly, in our experiments, Reparo-360 performs poorly under low bandwidth conditions, while LiveNAS-360 exhibits subpar performance when bandwidth is relatively abundant. This suggests that relying solely on single-domain neural enhancement is not able to adequately adapt to a wide range of bandwidth variations. In contrast, our approach considers the combination of both temporal and spatial domains, thereby possessing greater bandwidth adaptability and outperforming both baselines at all bitrate levels.

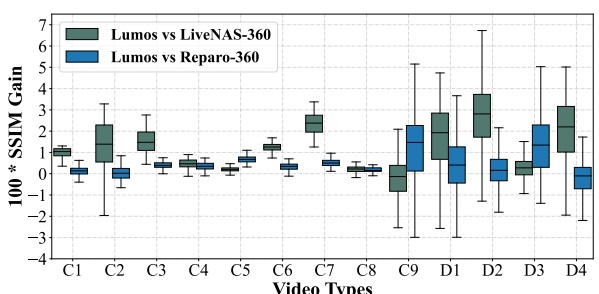

Figure 8: SSIM gain of video types.

Figure 7c and Figure 7d show the SSIM CDF curves for all video blocks on Gaze360 and SalientVR. We see that the majority of the Lumos curves lie below the other curves, which demonstrates the superiority of spatial-temporal integrated neural enhancement approach over those that only consider either the temporal or spatial domain. To further elucidate the effectiveness of Lumos, the chunk-level ssim gain compared to LiveNAS-360 and Reparo-360 on 13 types videos from the two datasets is shown in Figure 8, in which *C1-C9* refers to 9 video types in Gaze360 and *D1-D4* refers to 4 video types in SalientVR. Overall, Lumos has SSIM gain on the vast majority of video types, illustrating the robustness of our system.

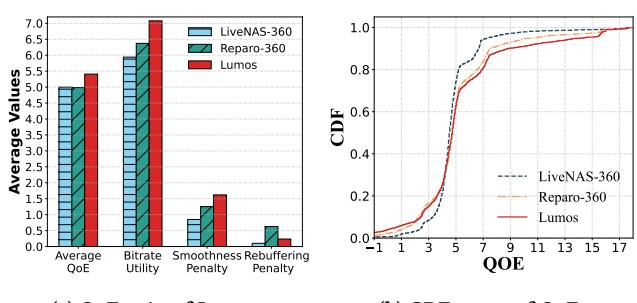

(a) QoE gain of Lumos.   (b) CDF curve of QoE.

Figure 9: QoE comparison of different methods.

**Quality of Experience.** To assess the QoE, we conducted tests on four types of videos from SalientVR using real network traces. Figure 9a illustrates the comparison of QoE between Lumos and two other baselines. The findings demonstrate that Lumos surpasses the other baselines, achieving the highest QoE, with an improvement of 8.2% on LiveNAS-360 and 8.5% on Reparo-360. To better understand QoE gains obtained by Lumos, we analyze its performance on each term for the QoE objective (see Eq. 2), which includes bitrate utility, smoothness penalty, and rebuffer penalty. For a clearer visualization, we multiply the rebuffer penalty by a factor of 20 while maintaining the original values for other components. As depicted in Figure 9a, Lumos exhibits the highest bitrate utility, with an increase of 19.1% on LiveNAS-360 and 11.1% on Reparo-360. Additionally, considering that Reparo-360 can discard no more than half of the frames which limits its adaptability to dynamic bandwidth, its rebuffer penalty is 2.67 times higher than that of Lumos. Since Lumos explores a significantly larger spatial-temporal neural enhancement configuration space, it suffers a little more smoothness penalty. Figure 9b presents the CDF curves of QoE for all video chunks of our four 360-degree videos. Overall, the majority curve of Lumos which corresponds to higher QoE value is below the other two baselines, suggesting that the spatial-temporal integrated neural enhancement paradigm is effective in improving QoE. Besides, there are small portions of the curve corresponding to lower QoE values above the other two baselines. This aligns with the smoothness penalty results shown in Figure 9a and will be addressed as part of our future work.

**Ablation Study.** To validate the effectiveness of the CBA and online training of the NQP model, we conducted an ablation study on two bitrate levels using four types of videos from SalientVR. Each video is sampled for 1 minute. As demonstrated in Table 1, online training of the NQP model leads to improvements in video quality at both bitrate levels, indicating the effectiveness of employing fragment samples for online training. Furthermore, substituting the CBA with the uniform distribution of bitrates hurts video quality. This underscores the necessity of dynamically allocating bitrates based on the spatial-temporal complexity of tiles, further emphasizing the effectiveness of the CBA.

*Case study of Content-aware Bitrate Allocator.* To further investigate the effectiveness of the CBA, we present the CDF curves of per-tile SSIM with and without CBA in Figure 10a. According to the

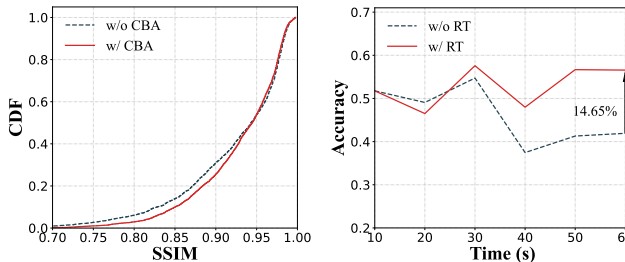

(a) CDF curves of per-tile SSIM w/ (b) Prediction accuracy of the NQP
and w/o CBA.       model w/ and w/o RT.

Figure 10: Case study of the Content-aware Bitrate Allocator and the online training of the NQP model.

results, employing CBA reduces the number of low-quality tiles while minimally affecting the quantity of high-quality tiles. This demonstrates the benefits of content-aware bitrate allocation based on spatial-temporal complexity.

*Case study of the online training of the NQP model.* Figure 10b displays the prediction accuracy of the NQP model with and without online training on video D2. We train the NQP model every 10 seconds and compare the SSIM ranking order of each neural enhancement configuration output by NQP with that on raw video frames. As depicted in figure 10b, over time, the retrained NQP model shows an improvement in accuracy of up to 14.65%, indicating the efficacy of our proposed online training method.

| Methods/Bitrates | 6Mpbs | 8Mpbs |
|---|---|---|
| w/o RT, w/o CBA | 0.9136 | 0.9275 |
| w/o RT, w/ CBA | 0.9215 | 0.9344 |
| w/ RT, w/o CBA | 0.9146 | 0.9284 |
| w/ RT, w/ CBA | **0.9233** | **0.9364** |

**Table 1: Ablation study of Lumos, where *CBA* denotes the Content-aware Bitrate Allocator, and *RT* denotes the online training of the NQP model using fragment samples.**

## 6 CONCLUSION

In this paper, we introduce Lumos, a novel live 360-degree video streaming system equipped with spatial and temporal integrated neural enhancement techniques to improve live 360-degree video quality under limited upload bandwidth. Lumos applies the real-time NQP model to forecast the enhanced quality for each neural enhancement configurations, dealing with the computational power bottleneck of upload client. The Content-aware Bitrate Allocator assigns bitrates to each tile and selects the appropriate neural enhancement configuration to improve overall 360-degree video quality. Besides, fragment samples are utilized to improve the prediction performance, preventing system performance from decreasing over time. Compared to the baselines using either spatial or temporal neural enhancement, Lumos achieves up to 0.022 SSIM gain and produces significant (8.2%-8.5%) QoE improvement for live viewers.

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
