# OpenReview forum: "Lumos: Optimizing Live 360-degree Video Upstreaming via Spatial-Temporal Integrated Neural Enhancement"
_acmmm.org/ACMMM/2024/Conference — MM2024 Poster_

### Official Review · Reviewer_diVL · 2024-05-21

**Rating:** 5
**Confidence:** 3

**Summary:**

The live-360 degree streaming has gained its popularity and has inbuilt faces the challenge to achieve right tradeoff between the QoE and uplink bandwidth. Neural enhancement with edge is best known solution that can save bandwidth in uplink and help client to avoid computational tasks, this method known to be used in single domain either spatial or temporal. The authors claim to introduce the combination of both for live 360-degree streaming. They use the prediction model to predict the neural enhanced quality for different video context  and design a context-aware bitrate allocator. The performance gain is significant with ~9% QoE improvement.

**Strengths:**

1. Live 360-degree streaming is challenging.
2. Using the combination of both spatial and temporal information to reduce bitrate and use the same for assigning higher bitrate is simple and effective.
3. Gain is good compared to SOTA.

**Limitations:**

1. I do not see any limitations

**Suitability:**

3

---

### Official Review · Reviewer_g2CJ · 2024-05-24

**Rating:** 4
**Confidence:** 3

**Summary:**

This paper introduced Lumos, a live 360-degree video streaming system equipped
with both spatial and temporal integrated neural enhancement. There is a
real-time neural-enhanced quality prediction model to forecast the enhanced
video quality, and a content-aware bitrate allocator to assign bitrates at the
tile level. Besides, fragment samples are utilized to improve the prediction performance.

**Strengths:**

* This paper is well written with very clear structure.

* The measurement study proved the drawback of one-dimension neural enhancement,
  which serves as a strong motivation for integrating both spatial and temporal
  enhencement.

**Limitations:**

* The QoE is not good. First, the QoE of Pensieve is proposed for 2-D video, not
  for 360-degree video. Second, the QoE metric does not reflect the latency,
  which is important in live streaming.

* SSIM is mainly used for images rather than video. Hence, other metrics like
  PSNR, VMAF should be considered.

* The evaluation results are not sufficient. As a live video solution, we
  are not only interested in the overall QoE improvement. Instead, the detailed
  improvement for every QoE component should be clearly illustrated. Moreover,
  it would be better to show the QoE improvement compared with spatial-only and
  temporal-only neural-enhancement.

**Suitability:**

3

---

### Official Review · Reviewer_m2Tw · 2024-05-25

**Rating:** 2
**Confidence:** 3

**Summary:**

This paper presents Lumos, a novel live streaming framework for 360-degree videos that integrates neural enhancement techniques in both the spatial and temporal domains. Lumos dynamically allocates bitrates using a Neural-enhanced Quality Prediction (NQP) model and a Content-aware Bitrate Allocator (CBA) to enhance video quality under constrained network bandwidth. Extensive experiments demonstrate that Lumos significantly improves the SSIM and QoE compared to the selected neural-enhanced baseline methods.

**Strengths:**

The main strength lies in the integration of spatial and temporal neural enhancement techniques into ive 360-degree video upstreaming, addressing the limitations of existing systems that focus primarily on only one domain.

The introduction of a lightweight, real-time NQP model enables the system to quickly predict the enhanced quality for various configurations, integrated with CBA that optimizing video quality efficiently without overburdening the upload client.

The authors provide an extensive evaluation using real-world network traces and diverse video datasets, demonstrating substantial improvements in SSIM and QoE over baseline methods.

**Limitations:**

The motivation behind the NQP model and online training is not well-explained. In measurement study section around figure 3, it is not clear how the relative orders of the upper and lower bounds of configurations relate to the feasibility of using low-quality training samples. Besides, it is unclear what advantages it offers compared to existing no-reference video quality estimation methods (e.g. VMAF or zero-inference method used in NeuralScaler). A comparative analysis in the experiments would be helpful.

The paper should better explain and exemplify (e.g., by visual samples) how the accuracy of the CBA and NQP affects output quality. Figure 10b shows that online training can improve prediction accuracy of the NQP but Table 1 shows that online update have a minor effects on quality, I'm not sure whether the effects to the visual experience really significant.

Although the system is designed for 360-degree video, it does not utilize much of the unique characteristics of 360-degree content. The paper should address why Lumos might outperform other advanced neural-enhanced live video methods, such as NeuralScaler, specifically for 360-degree video. Including this in the motivation and experimental comparison would strengthen the paper.

typo: In the "measurement study", there is a reference to "the lower bound (CRF33) of the configuration 540p, 12fps" in Figure 3, but Figure 3 does not contain a configuration for 540p, 12fps, CRF33. This inconsistency needs correction.

**Suitability:**

3

---

### Official Review · Reviewer_KZog · 2024-05-26

**Rating:** 3
**Confidence:** 3

**Summary:**

This paper presents a spatio-temporal neural-enhanced upstreaming framework for live 360-degree video.
Meanwhile, a neural-enhanced quality prediction model is developed to predict the neural-enhanced quality for different video content. However, the rationality of the system design needs further verification. The results are based on simulation evaluation.

**Strengths:**

In this work, the relationship between spatio-temporal complexity and tile rate distortion is explored and analyzed in detail. It also considers both spatial and temporal quality enhancement of neural networks. Furthermore, it is motivated by measurement experiments.

**Limitations:**

1) A single SR model is often difficult to adapt to various video contents. Therefore, in live streaming, it is often necessary to train SR models in real time. Proposed Neural-enhanced Quality Prediction (NQP) may be difficult to adapt to this situation.
2) Although the time cost of NQP per tile is 3ms, the bitrate allocation algorithm requires at least two inferences about each tile in each chunk, which is still very time consuming.
3) NQP only forecast the upper and lower bounds of the enhanced quality. It is not clear how to select the appropriate quality when there are many parameter configurations.
4) The bitrate allocation scheme based on the spatio-temporal complexity of the tiles may not be reasonable. For example, it may result in inconsistent quality between tiles in the same chunk?
5) The work controls the video quality by adjusting the CRF parameter, but the encoding bitrate of the tile can only be known after the it has been encoded, how can the bitrate assignment of tiles be done before encoding?
6) There are too many heuristics settings, such as weights pruning for bitrate assignment, considerdion of the highest and lowest CRF in NQP, and 10s buffer size. They may not work well for varying video content and real-world testing. Evaluations under simulation conditions have also shown limited user QoE gains, let alone in real-world systems.

**Suitability:**

3

---

### Meta-Review · Area_Chair_8ACq · 2024-06-27

**Recommendation:** Accept (Poster)
**Confidence:** 4

**Metareview:**

The paper integrates spatial and temporal neural enhancement techniques into 360-degree video upstreaming, addressing the limitations of systems that focus on a single domain. It introduces a lightweight, real-time NQP model that predicts enhanced quality efficiently and a CBA model that optimizes video quality without overburdening the upload client. Extensive evaluations using real-world network traces and diverse video datasets demonstrate significant improvements in SSIM and QoE over baseline methods. Some limitations remain, but several concerns have been addressed in the rebuttal. Regarding the quality of responses in the rebuttal and reviewers' feedback, I recommend accepting this paper as a poster.